# Laser Treatment as Sintering Process for Dispenser Printed Bismuth Telluride Based Paste

**DOI:** 10.3390/ma12203453

**Published:** 2019-10-22

**Authors:** Moritz Greifzu, Roman Tkachov, Lukas Stepien, Elena López, Frank Brückner, Christoph Leyens

**Affiliations:** 1Additive Manufacturing and Printing, Fraunhofer-Institut für Werkstoff- und Strahltechnik, Dresden 01277, Germany; roman.tkachov@iws.fraunhofer.de (R.T.); lukas.stepien@iws.fraunhofer.de (L.S.); elena.lopez@iws.fraunhofer.de (E.L.); frank.brueckner@iws.fraunhofer.de (F.B.); christoph.leyens@iws.fraunhofer.de (C.L.); 2Institute of Materials Science, Technische Universität Dresden, Dresden 01062, Germany; 3Department of Engineering Sciences and Mathematics, Luleå University of Technology, Luleå 97187, Sweden

**Keywords:** laser sintering, thermoelectric, bismuth telluride, antimony telluride, design of experiment, additive manufacturing

## Abstract

Laser sintering as a thermal post treatment method for dispenser printed p- and n-type bismuth telluride based thermoelectric paste materials was investigated. A high-power fiber laser (600 W, 1064 nm) was used in combination with a scanning system to achieve high processing speed. A Design of Experiment (DoE) approach was used to identify the most relevant processing parameters. Printed layers were laser treated with different process parameters and the achieved sheet resistance, electrical conductivity, and Seebeck coefficient are compared to tube furnace processed reference specimen. For p-type material, electrical conductivity of 22 S/cm was achieved, compared to 15 S/cm in tube furnace process. For n-type material, conductivity achieved by laser process was much lower (7 S/cm) compared to 88 S/cm in furnace process. Also, Seebeck coefficient decreases during laser processing (40–70 µV/K and −110 µV/K) compared to the oven process (251 µV/K and −142 µV/K) for p- and n-type material. DoE did not yet deliver a set of optimum processing parameters, but supports doubts about the applicability of area specific laser energy density as a single parameter to optimize laser sintering process.

## 1. Introduction

Thermoelectricity comprises several physical effects that occur in materials by means of voltage drop due to temperature gradient over the material (Seebeck effect) or development of temperature drops due to current flux (Peltier effect). These material properties are used in thermoelectric generators, e.g., as part of radioisotope generators to supply energy to deep space probes, when photovoltaic cells are no longer efficient. Probably the most familiar application for Peltier coolers is portable refrigerators. Currently, the most efficient materials for room temperature application are bismuth telluride and antimony telluride-based alloys. The standard route for producing generators/Peltier elements is compacting the material into disks, applying spark-plasma-sintering, cutting the disks into cubes, soldering them onto contacted Al_2_O_3_ substrates, lapping them, and soldering the top electrode. The approach is characterized by a considerable amount of manual work. An alternative approach to build up such devices is presented by additive manufacturing. Additive manufacturing describes a broad range of manufacturing technologies where material is added layer by layer to create a part. One of these processes is the SLM technique [1] (Selective Laser Melting), also called LPBF (Laser Powder Bed Fusion). The powder is spread homogeneously by a blade or a roll on a bed and a laser, coupled to a scan optic introducing energy into that spread powder, usually melting and consolidating it. Through repeating this process of spreading and laser processing, a 3D part is generated. This technology is usually used to process structural materials as stainless steel [2], aluminum [3], or titanium [4]. However, they have also already been used to build up different thermoelectric materials such as half-Heusler [5], Tin telluride [6], and bismuth telluride [7]. 

The feasibility of laser processing of Bi_2_Te_3_ powder compacts was shown in recent work of El-Desouky et al. [8]. Formation of the correct phases for the used parameters was confirmed by XRD measurements. However, there it is stated that commercially available Bi_2_Te_3_ powders usually do not fulfil the required specifications for additive manufacturing, especially for LPBF, which is also mentioned in Reference [7]. Two papers were published by a research group around Mao [9] and Wu [10]. Mao has synthesized Bi_2_Te_2.7_Se_0.3_ by self-propagating high-temperature synthesis (SHS). A slurry was produced, mixing the ground material with alcohol, and then the slurry was spread on a substrate. For comparison, a sample of the same powder was produced by conventional SPS (spark plasma sintering). It was found that the chemical composition of the material was dependent on the laser energy density. Energies higher than 33 J/mm³, especially tellurium and selenium, evaporated from the sample significantly. The strategy to compensate this effect was to enrich the starting composition of the compound, especially by using excess of tellurium. Good transport properties are measured, as shown in Table 1, though the properties are measured after 36 h of thermal annealing at 400 °C. The figure of merit (ZT) value at room temperature was calculated to 0.65. In Wu’s paper, the slurry is modified for dispenser printing. Also, DSC (differential scanning calorimetry) analysis is performed for as-processed samples in comparison to those which received thermal annealing at 400 °C for 6 h. These measurements indicate material inhomogeneity after LPBF, which is compensated by annealing. A bulk sample was produced by stacking 75 layers. The sample was used to measure the transport properties given in Table 1. Wu’s group has published on further experiments based on LPBF fabrication of thermoelectric materials like CoSb_2.85_Te_0.15_ [11], SnTe [6], or ZrNiSn [12].

Problems pointed out by References [7,8] in processing bismuth telluride powder by LPBF occur when trying to homogenously spread the powder, due to its arbitrary shape and broad particle size distribution. In additive manufacturing, spherical powders and narrow diameter distribution are usually favorable. Using the approach of creating a paste circumvents these issues, since for dispensing processes these perquisites to the material are less severe. Precisely, the overall idea is to use a dispenser for printing electrode, and p-/n-type thermoelectric materials in order to build up the whole thermoelectric device. The dispensing process is suitable for such an approach since multi-color printing is possible. The problem is seen in a thermal process to consolidate the printed materials and to adjust the thermoelectric properties.

In this work, laser processing is compared with more conventional furnace processing of bismuth telluride materials. The following Table 2 summarizes the results of the work, where thermoelectric materials were prepared as printable pastes and then thermally post treated in furnaces. It can be clearly seen that the range of achievable thermoelectric properties is broad. At a first glance, comparing the two tables, laser processing seems to lead to much better properties, but it should be noted that after laser processing, the authors of References [9,10] annealed their specimen for several hours at 400 °C, which exceeds most temperatures in Table 2. The achieved power factors shown in Table 2 do not exceed a power factor of 7.28 × 10^−4^ W/(mK²). Achieving high electrical conductivity seems the most critical issue, since Seebeck coefficients seem quite reasonable.

Compared to furnace heating, which usually takes hours, laser scanning is instead a very fast process and surfaces of some square millimeters can be scanned within less than a second. Laser treatment would allow the exact control of energy to be introduced into the printed layers, which could not be done in furnaces, but would be important, since e.g. p- and n-type materials have different sintering/melting temperatures. Laser processing would thus allow parallel build-up of both materials.

The thermoelectric community currently searches basically for new materials with high ZT values, since this increases the overall performance of Peltier or generator devices. Another focus is put on abundant or flexible materials [22].

From an engineering point of view, it is also very important to optimize and automatize the buildup of thermoelectric devices in order to decrease their cost, and thus for generators also the ratio of cost per generated power. Here, additive manufacturing presents a new and very promising alternative. 

## 2. Materials and Methods 

A paste was developed for the experiments conducted in this investigation, made of p-/n-doped bismuth telluride, polyvinylpyrrolidone (binder), and terpineol (solvent). The raw material was acquired as particles (mesh size 800, hence particles smaller than 15 µm) from Leshan Kai Yada Photoelectric Technology Co. Ltd (Hangzhou, China). Purity is specified as 99.99%. A check of the chemical composition of the materials by EDX (Energy-dispersive X-ray spectroscopy) resulted in the distribution, shown in Table 6. Subsequently, the selenium doped bismuth telluride is referred to as n-type and the antimony bismuth telluride as p-type. 

For printing, a dispenser printer (Musashi Engineering Europe GmbH., Muenchen, Germany) by Musashi, the Shotmaster 500 was used. Dispensing is a printing method, where a paste material is applied through pressure onto a substrate from a cartridge via a needle with a specific diameter. The cartridge is moved by the printer into x-, y-, and z-direction over the substrate by the printer. Parameters like printing velocity, applied pressure, needle diameter, and gap between needle and substrate influence the result and are usually optimized through a parameter study for each printing paste. Al_2_O_3_ substrates were used to print the layers on. The size of the printed layers was 7.5 × 7.5 mm or 10 × 10 mm.

A 600 W, 1064 nm, single mode, ytterbium fiber laser system with a beam parameter product (BPP) of 0.37 mm·mrad from the company IPG Laser GmbH (Burbach, Germany) was used. It was coupled to a 160 mm collimation (Coherent), a scanning system (IntelliScan20, Scanlab GmbH, Puchheim, Germany) and a 340 mm f–theta focusing lens (Sill Optics GmbH & Co. KG, Wendelstein, Germanz). All experiments were conducted out of the focal plane of the system, since working in the focus plane after first trials showed too-high energies which evaporated the printed material. The distance from focal plane was varied between 20 and 24 mm.

A portable mini-glove box was used with an optical window on the top for the laser beam. This box is placed under the scanner and flooded with argon to achieve inert gas atmosphere. The residual oxygen content was measured with an Oxy HP 2.0 system from OWT GmbH (Hagen am Teutoburger Wald, Germany). Experiments in Ar-atmosphere were conducted with less than 20 ppm oxygen.

Laser treatment was performed programming the laser-scanner system to emit a certain optical power *P* and moving the laser spot with the velocity *v* over the specimen’s surface. To treat a rectangular area, a hatch spacing *s*is defined. The laser then writes a line, switches off, is moved back to the starting point, shifted perpendicularly to the writing direction by the hatch spacing, switched on, and then writes a line again. This is repeated until the area is fully treated.

The tube furnace, which was used to produce reference specimen to be compared with the laser treated specimen is a furnace from HTM Reetz GmbH (Berlin, Germany). It can be connected to dry air, nitrogen, argon, or forming gas atmosphere. The tube oven can operate at maximum heat rates of 10 K/min, maximum temperature of 1150 °C, and does not provide active cooling. 

The characterization of the printed surfaces after thermal treatment was done by measuring thermoelectric properties. Sheet resistance was measured a by Keithley 2001 multimeter (HTM Reetz GmbH, Ohio, USA) in a four-point measurement setup. The setup is equipped with spring-loaded pins having 1 mm spacing, positioned in a line, and centered on the specimen to be measured. Measurements were conducted in ambient conditions. The following equation was applied to calculate electrical conductivity σ from the measured sheet resistance *R_sq_*_._:(1)$$\sigma ={\left(R_{sq.}\times w\times C\left(\frac{a}{d};\frac{d}{s}\right)\times F\left(\frac{w}{s}\right)\right)}^{-1}$$
with *w* – sheet thickness, and *C(a/d;d/s)* and *F(w/s)* being geometrical correction factors. In this investigation, due to used geometries, the factors were C(a/d;d/s) = 4.0095 and F(w/s) = 1 [23].

To calculate conductivity, the layer thickness has to be measured. This has been done using a micrometer screw gauge. Pictures were taken with a Keyence VHX5000 optical microscope (Keyence Germany GmbH, Neu-Isenburg, Germanz).

A commercial setup (SRX, Fraunhofer IPM, Freiburg, Germany) was used for the measurement of the Seebeck effect. All measurements were performed as stated in air or nitrogen. The applied current was kept below 10 mA in order to minimize joule heating effects. Seebeck coefficient measurements were carried out with temperature gradients of ca. 3 K. The sample was placed on two heaters. To set up a temperature gradient, one heater was activated while the other acted as the cold side. Temperature and voltage were measured by two thermocouples. Subsequently the triggering of the heaters was reversed so that the Seebeck coefficient was measured again. To ensure better electrical and thermal contact of the prepared thin films, carbon foil was applied. No further (sputtered) gold contacts were used whatsoever. The thermocouple distance for measuring the temperature and voltage was set to 3.0 mm. To ensure proper contact, U-I-curves were tracked and showed linear behavior.

The program Minitab (v18, Minitab GmbH, Muenchen, Germany) was used to create and conduct the DoE study. In a factorial design DoE study, a number “n” of parameters is fed to the related software. The program then responds with a set of experiments to be conducted, which can be envisioned as a regular n-dimensional body with n = ”number of parameters”. The edges of this body are represented by specific combinations of the parameter values. Since for this form of DoE, linear relation between experimental parameters and response is a precondition, a central point was included into the study. The software then creates a linear model of influence parameters and response, assigning factors to the influence parameters which describe their impact on the response. This model is expressed by a linear equation containing all chosen factors and all combinations thereof, describing these factors influence on the response parameter.

By stepwise backwards elimination of statistically non-relevant parameters, the model can be simplified and reduced to the relevant ones. The relevance of parameters can be seen in a Pareto Diagram. There, all bars crossing the red line are considered to have statistical significant influence on the response term. The red line is called the reference line and depends on the statistical significance level α. 

For DSC, equipment from Netzsch (Selb, Germany), namely the model DSC 404 F1 Pegasus was used. Experiments were conducted under argon atmosphere at heating and cooling rates of 10 K/min. Dry paste was put into Al_2_O_3_ pans, 21.41 mg for p-type and 37.66 mg for the n-type material.

Imaging and EDX measurements were carried out with the scanning electron microscope JEOL 7800 (JEOL GmbH, Freising, Germany) with an Oxford Xmax80 EDX detector. An element mapping was performed on a polished specimen, which was sputtered with carbon.

## 3. Results

In additive manufacturing, especially in the LPBF processes, the VED (volumetric energy density) is often used [9,24] as comprehensive parameter, even though it has been criticized as being too unspecific to describe the complex physical processes in the melt pool [24]. It is calculated from all relevant tunable parameters
(2)$$E_{V}=\frac{P}{v\times s\times h}$$
with *P* – laser power, *v* – scan velocity, *s* – hatch spacing, and *h* – thickness of layer. In contradiction to this, the area specific energy density *E_sq_*_._ was used in these experiments for comparing results of different laser process parameters with each other. It leaves out the layer thickness. The thickness of dispenser printed layers can actually be measured, but, anticipating results of this investigation, laser energy neither did propagate fully through the printed layers, nor was a clear border between treated and non-treated material visible. The area specific energy density *E_sq_*_._ is defined as
(3)$$E_{\mathrm{sq}.}=\frac{P}{v\times s}$$
with *P* – laser power, *v* – scan velocity, and *s* – hatch spacing. This parameter has been used, e.g., also by El-Desouky et al. [8].

A first set of parameters for laser processing was defined and experiments were conducted to approve the processing window. Also, processing in ambient atmosphere and Ar atmosphere was done, to find out the effect of the presence of oxygen on the process. One batch of specimens was printed for these experiments. Layer thickness was not characterized at this stage, but only the sheet resistance of the differently processed layers was measured. In parallel, a DSC analysis was made to find out the melting point of p- and n-type material. The results of the initial laser experiment are plotted as graph of sheet resistance versus *E*_sq._ in Figure 1.

It is observed that a clean Ar atmosphere leads to much lower sheet resistance of around 1 Ω/square, while processing in air leads to about 1000 Ω/square for low and 100 Ω/square for high *E*_sq._. Further, it appears that higher specific energy density leads to lower sheet resistance, especially in case of processing in air. Due to that result, all further experiments were conducted in Ar atmosphere.

Results of the differential scanning calorimetry of both materials are shown in Figure 2 and Figure 3. In the right part of the diagram for n-type, a clear melting point is visible at almost 580 °C. That peak was not seen in Reference [13] since the maximum temperature in that investigation was 450 °C. For the p-type material in the left part of the figure, the situation is more complex. Two peaks are visible, a first one at 410 °C and a second at 503 °C. Madan et. al. [13] found a peak for their p-type Sb_2_Te_3_ at 425 °C, commenting that this would be the melting temperature of Te or Te-rich phases. The first peak measured in this investigation probably corresponds to the one in Reference [13]. 

In a second step, using the results from the first laser experiments, the approach for optimizing the process was formalized by using design of experiment (DoE). Design of the study and its parameters are described in the subsequent paragraph. A new batch of specimens was produced by dispenser printing, including specimens for laser and tube furnace processing. At this step, sheet resistance and sheet thickness were measured in order to calculate electrical conductivity of the processed material. Also, the Seebeck coefficient was measured for some specimens. Cross sections were examined by optical microscopy as well as the top view of the laser treated layers.

The DoE study was performed for two different reasons. First, the previously done experiments raised doubts as to whether the integral parameter *E*_sq._ sufficiently describes the influence of laser radiation onto the sheet resistance of the treated layers. Secondly, an optimization of processing parameters is needed.

The DoE study was designed in such a way that not only the influence of the *E*_sq._ on the sheet resistance, but also the influence of its singular factors of laser power, scan velocity, hatch spacing (see Equation (3)), and additionally, the distance from the focal plane could be analyzed. The parameter set for the laser for the DoE study is shown in the Table 3. The same parameters were used for experiments with n- and p-type material, besides the laser power, where the maximum applied power was increased up to 150 W. This was done due to the higher melting temperature observed in DSC. The sheet resistance of the laser treated surfaces again was chosen as response parameter. 

Alternatively, the electrical conductivity could also have been chosen as the response parameter, but it was known from the first set of experiments that the laser parameters would have impact on both sheet resistance and sheet thickness, from which the conductivity is calculated. 

In the next paragraph, the results of the second set of experiments, including the DoE study, are presented. From a Pareto plot, the importance of an influence factor on a result is shown. The longer a beam in the diagram, the higher the influence of the corresponding parameter. 

For the p-type material, the Pareto plot in Figure 4a shows that all terms of Equation (3) significantly influence the sheet resistance in the order: scan velocity, hatch spacing, laser power, and combinations thereof. The distance from the focal plane came out to have no significant influence within the tested range. For the n-type material, the Pareto diagram (Figure 4b) looks more complex, indicating the hatch spacing to be the most relevant parameter, and also combinations of hatch–spacing–distance from focal plane, distance from focal plane–velocity, and a triple influence consisting of all parameters except distance from focal plane.

The measured sheet resistance *R*_sq._ was also plotted against *E*_sq._ in Figure 5. For the p-type material, the plot seems to show some potential decrease of resistance for growing laser energy, approaching a threshold around 1 Ω/square. For the n-type material, the relation appears to be linear. In the same diagram, the calculated electrical conductivity is plotted, which will be discussed later. 

To create benchmark results for furnace treated specimens, to be compared with the laser treated ones, three specimens of each material were processed in a tube furnace for 30 min at 350 °C in forming gas (4% H_2_/96% N_2_). Thermoelectric properties of these samples are shown in Table 4. Here, the thermoelectric properties are among the good ones compared to results from other studies, which are shown in Table 2. For p-type material, laser processing produced several samples of sheet resistance equal or lower than the furnace processed samples. For the n-type material, this is absolutely not the case, since the lowest sheet resistivity is measured as 2 Ω/square at 0.2 J/mm².

The thickness of the layers was measured with a micrometer screw gauge. The graph in Figure 6 shows the normalized sheet thickness of the laser treated layers. The sheet thickness of the furnace specimens was used as a reference value. Using a normalized value in the graph was set, since all specimens were from the same batch, and thus contained the same amount of conductive material, and the pre-treatment thickness can be considered as equal. It can be seen that especially for high values of *E*_sq._, the thickness is growing up to almost the double of an oven-processed layer. According to Equation (1), increasing sheet thickness would decrease conductivity σ, which is obviously just true for bulk material.

Also, cross sections (Figure 7) were prepared from p-type laser treated samples for three different *E*_sq._. There, it is clearly visible that not the whole layer was molten, but only a thin layer on the top. Also, no smooth surface is visible, but a disturbed one with a high roughness.

In Figure 8a–f, top view images of the laser treated surfaces are seen, stating the applied energy density in the caption. Figure 8a–c shows the development for increasing *E*_sq._ for p-type material. Here already at low temperatures, a kind of clustering of material can be observed, with droplet-like objects appearing between 20–50 µm. For higher energies, the buildup of bridges between these droplets can be seen. At the highest *E*_sq._, big bubbles are seen with a dimension of more than 200 µm, with shiny metallic surfaces, while the rest of the surface appears dim with network like structures. Figure 8d–f shows the development for the n-type material. Here, the development seems to be different. Figure 8d looks like an as-printed, dried surface, with some visible, slightly darker lines. For higher energies, bubble structures appear, that grow larger at higher energy. Bridges or network formation is not observed.

Finally, the Seebeck coefficient was measured for three different *E*_sq._ for each material, 0.07, 0.10, and 0.13 J/mm² for p-type and 0.15, 0.20, and 0.25 J/mm² for n-type, respectively. The obtained Seebeck coefficients were much lower than for the furnace samples (shown in Table 4). Further, for n-type material almost no deviation is seen for different *E*_sq._. For p-type, better Seebeck coefficient Figure 9 is measured for lower *E*_sq._. For n-type, approximately 77% of the Seebeck coefficient of furnace treated samples can be measured, while for p-type, the Seebeck coefficient decreases to 16–28%.

EDX (Energy-dispersive X-ray spectroscopy) was performed in order to check for deviations of the elemental composition. Figure 10 shows the area under investigation. It is a section from the same cross cut as shown in Figure 7b treated with *E*_sq._ = 0.154 J/mm². Area 1 clearly lies in the molten section, while Areas 2 and 3 are used as references since no molten particles can be identified in them. Areas 4 and 5 were slightly highlighted during EDX analysis for having higher tellurium content and were thus separately scanned. 

The results of the elemental composition are shown in Table 5 as a comparison of a furnace treated sample and the different areas of the laser treated sample. Areas 2 and 3 are considered to be equal, since no deviation larger than 0.1% is seen. Between the oven sample and Areas 2 and 3, there are slight deviations not exceeding 1.5%. Two percent less Sb is found in the molten Area 1, but Bi and Te also do not deviate more than 1%. Areas 4 and 5 deviate remarkably, since there is almost no Bi. The 4.7 wt.% measured in Area 4 are near the detection limit. On the other hand, in these areas, the fraction of Sb is very high. Not seen in the table are the high amounts of oxygen and carbon, also found in Areas 4 and 5. Figure 11 shows the measured peaks for Area 5, where these peaks are clearly visible. The chemical composition of the purchased material is shown in Table 6.

## 4. Discussion

According to the results shown in Figure 1, laser processing of bismuth telluride-based materials should be better done in inert atmosphere. This is actually done by the researchers of all publications found on laser processing of such kind of materials [9,10]. Nonetheless, it is notable that processing in ambient air results in conductive layers at all, since furnace processing in ambient air leads to fully oxidized layers. That a resistance is measurable at all is suspected to be possible due to the much shorter processing times and thus much more limitation time for oxidation processes. 

Higher laser energies (80–120 W), even if a defocused laser is used, do not lead to the desired faster process in comparison to the work in References [9,10] where 3–10 W equipment is used. The intensity and *E*_sq._ of laser radiation and processing parameters is comparable, but still, the layers fabricated here were not homogeneously molten. This is probably due to the much faster scan speeds and thus due to less time in which each segment of a layer is irradiated, so that energy cannot propagate into the depth of the printed layers. 

On the other hand, in the two quoted papers, after laser processing, 6 h or even 24 h of annealing at 400 °C was performed. That temperature already presents 70% of the endothermic peak found in Reference [13] and in this work, and there might be already sintering activities. So, nothing was known about the thermoelectric properties of as laser treated layers, yet. The finding that VED is not a sufficient parameter to describe results of laser sintering or melting processes for additive manufacturing was published in Reference [24]. The results of the here presented investigation would support that assumption, extending it to area specific density and thermoelectric parameters as properties to be achieved. In Figure 5, Figure 6, and Figure 9, where sheet resistance, sheet thickness and Seebeck coefficient are shown, it is visible, that for same value of *E*_sq._ different of the just named properties are measured.

N- and p-type materials reacted quite differently to the laser treatment and also statistical analysis via DoE resulted in very different influence parameters. For p-type, the statistical model seems to be comparatively clear and includes all parameters, which are also part of Equation (3) to calculated *E*_sq._. Nonetheless, it is shown that scan velocity has a higher influence on the sheet resistance than power or hatch spacing. Unfortunately, the Pareto diagram for the n-type material looks much less clear for the tested parameters. The structures in Figure 8d–f, where top views of the n-type laser treated specimen are shown, remain on balling effects that are encountered for too high scan speeds in LPBF. If the findings in Reference [24] for LPBF are applicable to the process and materials presented here, slower scan speed at lower laser power at the same *E_sq_*_._ might lead to better results. However, the findings in Reference [24] might not be directly applicable. In LPBF, metals are irradiated, while in the work presented here, they are composites of metals and organic compounds. Another reason for the archived results could also be that either energy propagation in composites is poor due to few contacts between metal particles, or also due to evaporating organics.

Inhomogeneity and increased sheet thickness of the laser processed layers presents a major problem when characterizing electrical conductivity. That parameter does not depend on geometry and thus would have allowed better comparison to results, achieved by other groups. Since for the furnace and laser experiments, the same batch of printed specimens was used, at least for the work presented here the results of the sheet resistance are comparable. 

Decrease of Seebeck effect in the shown samples is matter of discussion. The depression of Seebeck due to better conductivity is unlikely, since especially for n-type, the conductivity is worse for laser process compared to furnace treatment and for p-type, the conductivity is still in the same range. Also, for p-type, the samples that were treated at modest *E_sq_*_._ show higher Seebeck coefficient. EDX analysis was performed to check if the chemical composition was changed during laser processing, e.g., due to evaporation of elements as indicated in Reference [10]. This was not precisely found in the analysis as the chemical composition was in principle similar to furnace-treated samples. Only in some tiny droplets on the surface of the scanned specimen was there a completely different composition, where almost no Bi is found, but the mass fraction of Sb is much higher. The high oxygen and carbon contents on the surface are not quantifiable by this method. The peaks are too high to be attributed to cross section preparation or EDX measurement itself, but on the other hand, it is assumed that here condensed organic material that was evaporated during laser process is seen. As for the Seebeck effect, a change of charge carrier density is suspected to happen due to segregation of elements, which would not be detected by EDX.

## 5. Conclusions

Laser processing as thermal treatment was performed on n- and p-type bismuth telluride-based materials. It was shown that such treatment should be performed under inert atmosphere. Optimum processing parameters were not found for both materials, but it was shown, even though the materials can be considered similar, that they still need different treatment parameters. Material is molten on the surface of the printed layers, while in lower regions the particles appear as printed. DoE can be an approach to optimize these parameters, since the area specific laser energy density *E*_sq._ does not sufficiently describe the process. Higher laser power does not necessarily allow faster processing speed. DoE should be applied to processing windows with lower power and scan speeds, where in previous studies good results were found. Electrical conductivity was found to be slightly better (p-type) or worse (n-type) for laser treatment compared to furnace processing. The depression of Seebeck effect in as-laser treated specimen remains subject of investigation. The material properties are thus in a similar range for laser and furnace processing. Nonetheless, laser treatment is considered to be a promising approach due to its high processing speeds in comparison to furnace processing. Also, as shown, melting of the used material is induced by the laser which is not the case for the furnace process, where even sintering has not been found. Melting or sintering should actually enhance the electrical conductivity. Also, using a laser for processing the materials on temperature sensitive substrates is more likely than using a furnace.

## Figures and Tables

**Figure 1 materials-12-03453-f001:**
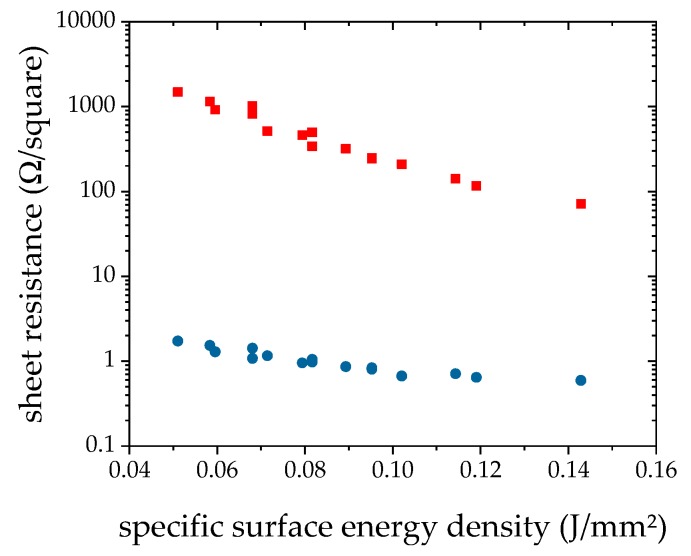
Comparison of sheet resistance of n-type bismuth telluride when processed in air (red squares) and Ar (<20 ppm O_2_-content < 20 ppm; blue dots) atmosphere. Power P = 100 W, Scan velocity v = 2.5–4 m/s, spot diameter d = 562 µm, hatch spacing h = 0.3–0.5 mm.

**Figure 2 materials-12-03453-f002:**
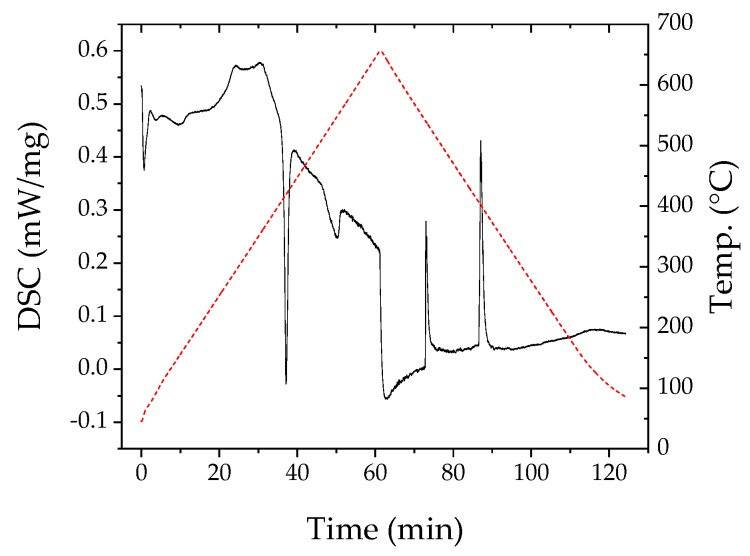
Differential scanning calorimetry (DSC) results for dry p-type paste (red dotted line shows temperature curve).

**Figure 3 materials-12-03453-f003:**
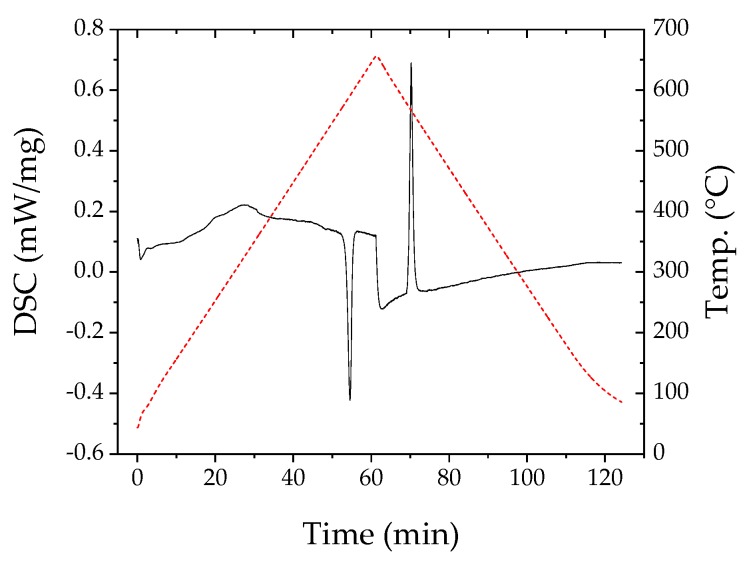
Differential scanning calorimetry (DSC) results for dry n-type paste (red dotted line shows temperature curve).

**Figure 4 materials-12-03453-f004:**
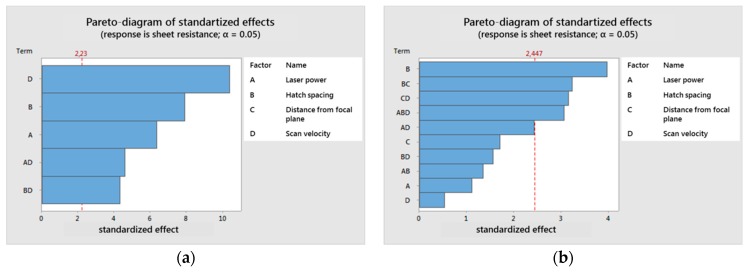
Pareto plot for the reduced statistical models calculated by Minitab, (**a**) p-type; (**b**) n-type.

**Figure 5 materials-12-03453-f005:**
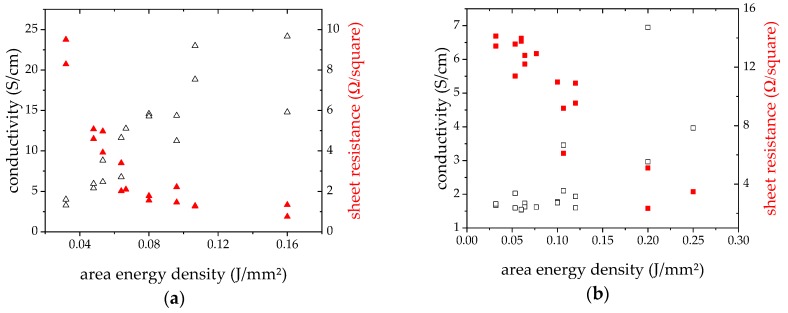
Sheet resistance and conductivity vs. *E*_sq._, (**a**) p-type; (**b**) n-type (empty symbols correlated to conductivity, red filled symbols to sheet resistance).

**Figure 6 materials-12-03453-f006:**
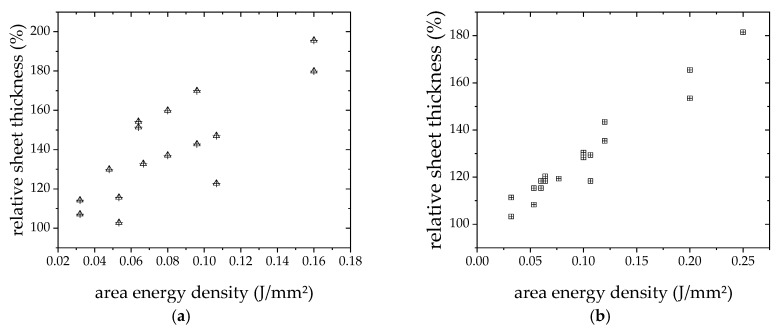
Sheet thickness vs. *E_sq_*_._ (**a**) p-type; (**b**) n-type.

**Figure 7 materials-12-03453-f007:**
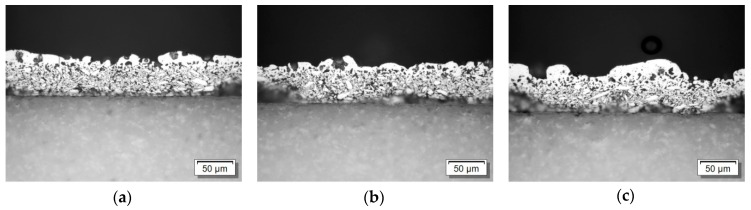
Cross section of laser treated specimen: (**a**) *E*_sq._ = 0.088 J/mm²; (**b**) *E*_sq._ = 0.154 J/mm²; (**c**) *E*_sq._ = 0.206 J/mm².

**Figure 8 materials-12-03453-f008:**
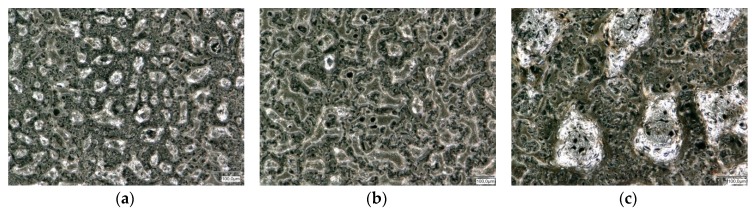
Top view microscopy images of p-type material treated with (**a**) *E*_sq._ = 0.048 J/mm²; (**b**) *E*_sq._ = 0.080 J/mm²; (**c**) *E*_sq._ = 0.160 J/mm²; and n-type material treated with (**d**) *E*_sq._ = 0.053 J/mm²; (**e**) *E*_sq._ = 0.107 J/mm²; (**f**) *E*_sq._ = 0.200 J/mm².

**Figure 9 materials-12-03453-f009:**
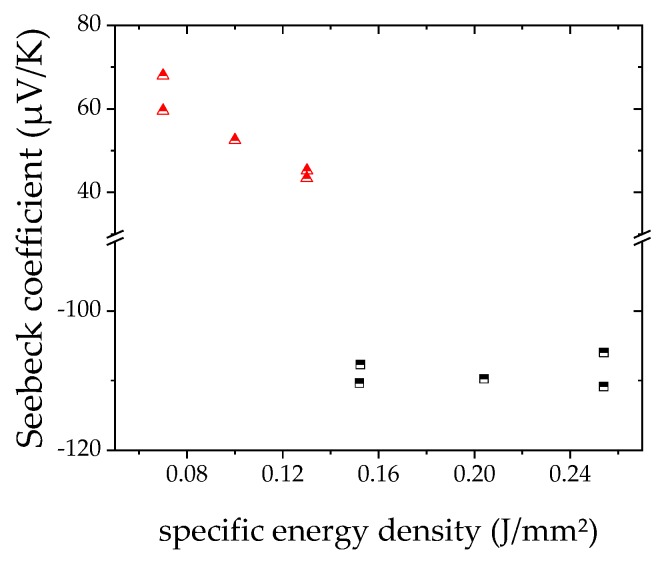
Seebeck coefficient for laser treated bismuth telluride layers (red triangles p-type; black squares n-type).

**Figure 10 materials-12-03453-f010:**
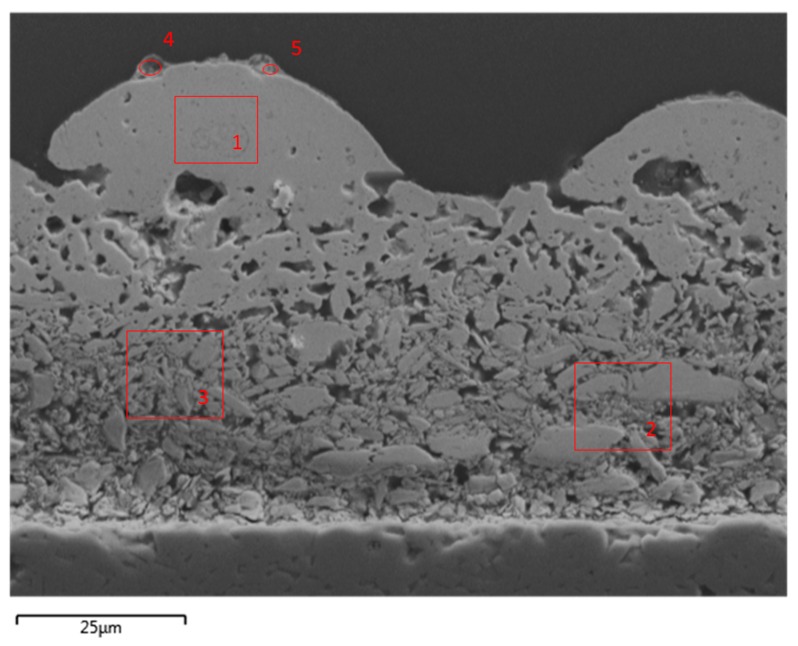
Scanning electron microscopy (SEM) image of area under investigation for energy-dispersive X-ray spectroscopy (EDX) with 5 areas, where elemental composition was analyzed.

**Figure 11 materials-12-03453-f011:**
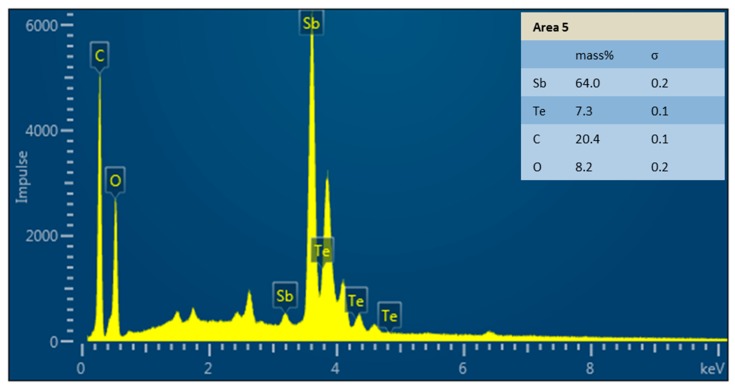
Peaks of the energy-dispersive x-ray spectroscopy (EDX) analysis of Area 5.

**Table 1 materials-12-03453-t001:** Overview over previous studies on laser treatment of bismuth telluride materials.

Ref.	Material	Wavelength (nm)	*E*_sq._ (J/mm²)	Seebeck Coefficient (µV/K)	Electrical Conductivity (S/cm)	Power Factor (W/mK²)
[8]	Bi_2_Te_3_ (powder)	1070	0.6–1.4	Not measured	Not measured	Not measured
[7]	Bi_2_Te_3_ (powder)	1070	1.3	−166	600	1.65 × 10^−3^
[9]	Bi_2_Te_2.7_Se_0.3_ (dried slurry)	1064	0.3–11 ^1^	−145	700	1.47 × 10^−3^
[10]	Bi_2_Te_2.7_Se_0.3_ (printable paste)	1064	0.5–1.5 ^1^	−105.7	122	1.36 × 10^−3^

^1^ calculated from volumetric density that was given in the papers.

**Table 2 materials-12-03453-t002:** Overview of previous work on bismuth telluride materials that were thermally post treated in furnaces.

Ref.	Thermal Processing	Atmosphere	Material	Seebeck Coefficient (µV/K)	Electrical Conductivity (S/cm)	Power Factor (W/mK²)
[13]	12 h @ 350 °C	n.a.	Sb_2_Te_3_	152	315	7.28 × 10^−4^
	48 h @ 350 °C	n.a.	Bi_2_Te_3_	−287	14	1.15 × 10^−4^
[14]	12 h @ 250 °C	Ar	Bi_2_Te_3_ (+2% Se)	−190	40	1.44 × 10^−4^
[15]	12 h @ 350 °C	n.a.	Bi_2_Te_3_ (+1% Se)	−170	100	2.89 × 10^−4^
[16]	12 h @ 250 °C	n.a.	Bi_0.5_Sb_1.5_Te_3_ (+8% Te)	280	23	1.80 × 10^−4^
[17]	12 h @250 °C	Ar	Bi_0.5_Sb_1.5_Te_3_ (+8% Te)	250	11	6.88 × 10^−5^
			Bi	−84	110	7.76 × 10^−5^
[18]	3 h @ 250 °C	n.a.	Sb_2_Te_3_	137	53	9.88 × 10^−5^
			Bi_1.8_Te_3.2_	−132	4	6.97 × 10^−6^
[19]	3 h @ 250 °C	n.a.	Sb_2_Te_3_	103	200	2.12 × 10^−4^
			Bi_1.8_Te_3.2_	−145	7	1.40 × 10^−5^
[20]	3 h @ 250 °C	N_2_	Sb_2_Te_3_	109	278	3.27 × 10^−4^
			Bi_1.8_Te_3.2_	−138	100	1.92 × 10^−4^
[21]	10 min @ 500 °C	Air	ZnSbO	225	30	1.52 × 10^−4^
			CoSb_3_	−49	211	5.05 × 10^−5^

**Table 3 materials-12-03453-t003:** Parameter matrix for Design of Experiment (DoE) factorial design.

Material	p-Type		n-Type	
	min	max	min	max
Parameter	−1	+1	−1	+1
Laser power (W)	80	120	80	150
Scan velocity (m/s)	2.5	5.0	2.5	5.0
Hatch spacing (mm)	0.3	0.5	0.3	0.5
Distance from focal plane (mm)	20	24	20	24

**Table 4 materials-12-03453-t004:** Electrical characterization of furnace processed layers.

Material	Conductivity (S/cm)	Sheet Resistance (Ω/square)	Thickness (mm)	Seebeck Coefficient (µV/K)	Power Factor (W/mK²)
p-type	15.3 ± 1.5	2.4 ± 0.2	70.0 ± 3.0	251	9.64 ×·10^−5^
n-type	88.0 ± 3.7	0.3 ± 0.1	99.7 ± 1.3	−142	1.77 ×·10^−4^

**Table 5 materials-12-03453-t005:** Mass fraction in % of the elements of p-type material after oven processing and in several areas of a laser processed specimen (*E*_sq._ = 0.154 J/mm²).

Element	Oven Processed	Area 1 (Near Surface)	Area 2 (Bottom)	Area 3 (Bottom)	Area 4 (Detail Surface)	Area 5 (Detail Surface)
Te	56.1%	58.8%	57.6%	57.7%	22.6%	7.3%
Sb	27.8%	25.0%	27.2%	27.3%	72.7%	64.0%
Bi	16.1%	16.1%	15.2%	15.1%	4.7%	-

**Table 6 materials-12-03453-t006:** Results of energy-dispersive X-ray spectroscopy (EDX) analysis, values in wt.%.

Element	n-type	p-type
Bismuth	53.7	16.1
Tellurium	43.9	56.1
Selenium	2.4	-
Antimony	-	27.8

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
