# Peer review of "Laser Treatment as Sintering Process for Dispenser Printed Bismuth Telluride Based Paste"

_materials, 2019, doi:10.3390/ma12203453_

Round 1
Reviewer 1 Report
The manuscript can be accepted as it isThe manuscript can be accepted as it is.What countermeasures do the author plan to further reduce oxygen and/or porosity?
Author Response
Dear Reviewer,
thank you very much for your time and effort.
We have created a new version of the draft containing the comments and changes of all reviewers and the academic editor.
Please see the attachment

Reviewer 2 Report
This work is a systematic study using the DoE of laser sintering for printed p and n type thermoelectric paste. The overall results were presented clearly and systematically, however the findings lacked significance.
Author Response

(The authors gave the same response as above.)

Reviewer 3 Report
Other laser types (different power and/or wavelength) could be studied in order to find the most suitable ones, with a new set of conclusions, some of them maybe more positives.
Line 99 - Table 3 could be moved to the next page;
Line 366 - References could be moved to the next page;
General layout of the paper could be slightly improved.
Author Response

(The authors gave the same response as above.)

Reviewer 4 Report
This paper investigates the dispenser printed p- and n-type bismuth telluride based thermoelectric paste materials by using a laser sintering as a thermal post treatment method. Some material properties were demonstrated in comparison with furnace tube processing. Although they did not find the optimal processing parameters, their presented results were valuable for some researchers. I suggested that this paper can be accepted for publication in this journal.
Author Response

(The authors gave the same response as above.)

Reviewer 5 Report
Content of the manuscript is appropriate for the audience of the Journal and the reviewer has some suggestions and recommendations to improve the manuscript.Therefore, the reviewer recommends that the manuscript is worthy for publication with the following corrections:
General comment:
Please check some typos in the paper and uniform the symbols of numerical values (e.g. line 79 pag. 2: 7.28*10-04 W/(mK²) or other values of Power factor in Table 2.) Please remove * and change with another symbol ( ). ·
Please check all the equations. For example, Eq 1 and 3 have some wrong things in the text.
In Figures 1,2,3,5, and 9 the all axes are not reported. Please use the same format for all figures. Authors are requested to increase the font size and also the thickness of lines and axes for better readability and quality.
Please emphasize better your work because it’s not clear if the application of laser sintering can or not improve the properties of the material. If the materials fabricated with this technique are more or less similar to materials prepared with traditional approach, the authors have to clarify why we should apply the laser sintering. Add some comments related to the advantages of this technique (save time? Low-cost starting materials? Better tailoring of other properties?. Please clarify it.
Abstract: Remove the sentence “This investigation contributes to the topic of additive manufacturing processes for the build-up of thermoelectric devices”
Introduction I suggest the authors to focus the introduction of the main aim of the paper. In fact, it’s not clear from the introduction the benefits of this research. The main weak points of the submitted paper are probably due to the poor bibliography (only 18 references). Indeed, the authors do not seem to have a good idea of the advances in the various modern improvements in the laser sintering technology that is a consolidated technology.
I suggest the authors to improve the introduction with more references and with a comparison with relevant literature.
Materials and methods:
The sentence “A paste was developed for the experiments described here “is unclear because there are no references. In addition, considering that the laser sintering seems the main point of the paper the authors have to report the analytical procedure to perform it.
Table 3 has to be reported in the results.
Add in materials and method section the name and model of all equipment used for characterization (e.g. name and company of SEM microscope etc..)
The equipment for laser sintering has to be described better in this section.
Results
Figures 2 and 3 in this form are very confused. Please redraw these Figures.
Author Response

(The authors gave the same response as above.)

Round 2
Reviewer 5 Report
The authors modified the paper according to most of my suggestion, however in order to improve the readability of the paper I ask to the authors do modify these points:
1) according to my previous revision all figure need the same format.
2) Add top axis in Figures 2,3 and 5. Add top axis and right axis in Figure 1, 6 and 9.